# A Label-Free Electrical Impedance Spectroscopy for Detection of Clusters of Extracellular Vesicles Based on Their Unique Dielectric Properties

**DOI:** 10.3390/bios12020104

**Published:** 2022-02-09

**Authors:** Yuqian Zhang, Kazutoshi Murakami, Vishnupriya J. Borra, Mehmet Ozgun Ozen, Utkan Demirci, Takahisa Nakamura, Leyla Esfandiari

**Affiliations:** 1Department of Surgery, Division of Surgical Research, Mayo Clinic, Rochester, MN 55905, USA; zhang.yuqian@mayo.edu; 2Microbiome Program, Center for Individualized Medicine, Mayo Clinic, Rochester, MN 55905, USA; 3Division of Endocrinology, Cincinnati Children’s Hospital Medical Center, Cincinnati, OH 45229, USA; kazumurakami@okayama-u.ac.jp (K.M.); vishnupriya.borra@cchmc.org (V.J.B.); takahisa.nakamura@cchmc.org (T.N.); 4Canary Center at Stanford for Cancer Early Detection, Department of Radiology, Stanford School of Medicine, Stanford University, Palo Alto, CA 94305, USA; mehmeto@stanford.edu (M.O.O.); utkan@stanford.edu (U.D.); 5Bio-Acoustic MEMS in Medicine (BAMM) Laboratory, Department of Radiology, School of Medicine, Stanford University, Palo Alto, CA 94304, USA; 6Department of Pediatrics, College of Medicine, University of Cincinnati, Cincinnati, OH 45221, USA; 7Department of Metabolic Bioregulation, Institute of Development, Aging and Cancer, Tohoku University, Sendai 980-8577, Miyagi, Japan; 8Department of Biomedical Engineering, College of Engineering and Applied Sciences, University of Cincinnati, Cincinnati, OH 45221, USA; 9Department of Electrical Engineering and Computer Science, College of Engineering and Applied Sciences, University of Cincinnati, Cincinnati, OH 45221, USA; 10Department of Environmental and Public Health Sciences, University of Cincinnati, Cincinnati, OH 45221, USA

**Keywords:** extracellular vesicles (EVs), exosome, dielectric properties, electrical impedance spectroscopy (EIS), insulator-based dielectrophoretic (iDEP), biosensor

## Abstract

Extracellular vesicles (EVs) have gained considerable attention as vital circulating biomarkers since their structure and composition resemble the originating cells. The investigation of EVs’ biochemical and biophysical properties is of great importance to map them to their parental cells and to better understand their functionalities. In this study, a novel frequency-dependent impedance measurement system has been developed to characterize EVs based on their unique dielectric properties. The system is composed of an insulator-based dielectrophoretic (iDEP) device to entrap and immobilize a cluster of vesicles followed by utilizing electrical impedance spectroscopy (EIS) to measure their impedance at a wide frequency spectrum, aiming to analyze both their membrane and cytosolic charge-dependent contents. The EIS was initially utilized to detect nano-size vesicles with different biochemical compositions, including liposomes synthesized with different lipid compositions, as well as EVs and lipoproteins with similar biophysical properties but dissimilar biochemical properties. Moreover, EVs derived from the same parental cells but treated with different culture conditions were characterized to investigate the correlation of impedance changes with biochemical properties and functionality in terms of pro-inflammatory responses. The system also showed the ability to discriminate between EVs derived from different cellular origins as well as among size-sorted EVs harbored from the same cellular origin. This proof-of-concept approach is the first step towards utilizing EIS as a label-free, non-invasive, and rapid sensor for detection and characterization of pathogenic EVs and other nanovesicles in the future.

## 1. Introduction

Extracellular vesicles (EVs), including exosomes (40–150 nm) and microvesicles are released from many cell types into extracellular spaces and are circulated in almost all biofluids, including blood, urine, breast milk, cerebral fluids, and saliva [1]. They are taken up by neighboring or distant cells and subsequently modulate functions of the recipient cells. EVs are composed of a lipid bilayer membrane containing unique receptors and tetraspanin surface markers. They also encapsulate exclusive cargos in their lumen, including proteins, lipids, and nucleic acids [2]. The unique composition of EVs reflects their parental cells with both physiological and pathological relevance [3]. Thus, detection and characterization of EV surface markers and cargos offers great opportunity for early diagnosis and monitoring the prognosis of several diseases, including cancer, cardiovascular disease, and degenerative disorders [4]. The state-of-the-art technologies are mainly based on EVs’ biophysical characterization, including their size distribution, density, and morphology, and can be listed as transmission electron microscopy (TEM) [5], nanoparticle tracking analysis (NTA) [6], and density gradient separation [7]. However, these techniques are either low throughput and time-consuming to operate or do not provide information with regards to EV’s biochemical properties, cellular origins, and functionality. Thus, in recent years, flow cytometry has been adopted as a high-throughput method for characterization of EVs based on their biochemical properties by labeling their specific protein markers, membrane lipids, or nucleic acids [8]. Although flow cytometry has shown promising attributes, it is a label-based technique which relies on the specificity of antibodies to the targeted receptors. More importantly, flow cytometry lacks accuracy for characterization of EVs with a smaller size distribution, since the scatter sensitivity of current technologies is limited to EVs larger than ~100 nm [9]. Other analytical methods, such as western blot, mass-spectrometry (MS), microarray technology, and RNA sequencing, are applied to study the abundance of EVs’ proteins, lipids, and nucleic acids [10]. Although these techniques are highly sensitive for EVs’ biochemical profiling, they require lysis or labeling steps prior to screening, which not only add time and cost to the procedure, but also break the structure of the vesicles. Considering the therapeutic potential of EVs, it is important to maintain EVs’ intact structure and native composition.

Electrical impedance spectroscopy (EIS) is a label-free and non-invasive technology that has been developed for measuring the impedance of cells under an alternating current (AC) over a wide range of frequencies, aiming to characterize their dielectric properties which resemble their unique membrane and cytosolic compositions [11,12,13]. This technique has been widely utilized to differentiate stem cells [14,15] and cancerous cells [16,17]. In the majority of EIS techniques, a single cell is initially trapped at a fixed position, followed by impedance measurement of the cell at a selected frequency range [18]. The variation of the impedance signal provides information on cells’ morphological and electrophysiological changes which are related to the cells’ intrinsic dielectric properties. Microfluidic flow cytometry (MFC) is another impedance-based cellular analysis, in which a single cell dynamically flows through a channel with embedded micro-electrodes. The impedance of a cell at a wide frequency spectrum is collected for the analysis of its properties, including size, membrane capacitance, and cytoplasmic conductance [19,20]. However, the application of EIS tools for detection of EVs with heterogeneous and nanoscale-size distributions has not been explored.

In this proof-of-concept study, we have adopted EIS to detect a cluster of EVs harvested from different cellular origins and investigated the correlation between their impedance responses and their intrinsic dielectric properties, including their unique membrane and cytosolic characteristics. Although in principle a single vesicle detection would provide important information with regards to its biochemical composition, similar to a single cell analysis by EIS, it would be extremely challenging to develop a high throughput device to cover the heterogeneous size distribution of exosomes (40–150 nm) with high resolution [21,22]. Thus, in this initial study, we focus on the detection and characterization of clusters of EVs collected from different parental cells or culture conditions in a high throughput manner. In this system, EVs were first immobilized by an iDEP device developed by our team [23,24], followed by sweeping of an AC field at 100 mV_rms_ from 0.5 MHz to 50 MHz for impedance measurements by an integrated EIS, as illustrated in Figure 1a. We initially detected liposomes and carboxylic acid polystyrene (COOH-PS) beads of similar size with known dielectric properties and followed this with the construction of an equivalent circuit model for theoretical validation. The system has further been utilized to characterize different EVs with different membrane properties as well as to treat them with different stimuli in culture (Figure 1(bi)) to obtain the impedance responses to variation in their membrane and cytosolic dielectrics at a wide range of frequencies. Moreover, we utilized the system to differentiate EVs from lipoproteins and detect EVs derived from different cellular origins (Figure 1(bii)) and EVs secreted from the same cellular origin but with different size ranges (Figure 1(biii)). Overall, this approach established a rapid and label-free detection scheme for characterization of EVs with different biochemical compositions and potentially functionality, laying a foundation to leverage EVs as circulating biomarkers for disease diagnosis and prognosis or as personalized therapeutic cargos.

## 2. Materials and Methods

### 2.1. Materials

All chemicals were purchased from Sigma-Aldrich (St. Louis, MO, USA) unless otherwise noted. Silicone elastomer base and curing agents were purchased from Dow Corning (Elizabethtown, KY, USA). Platinum electrodes were purchased from Alfa Aesar (Haverhill, MA, USA). Phosphate-buffered saline (PBS) was purchased from Roche Diagnostics (Indianapolis, IN, USA). Borosilicate pipettes with filament (O.D. 1 mm; I.D. 0.78 mm; length 7.5 cm) were obtained from Sutter Instrument (Novato, CA, USA). One-hundred nanometer liposomes (phospholipid DOPC and cholesterol) were purchased from FormuMax Scientific, Inc. (Sunnyvale, CA, USA). One-hundred nanometer COOH-PS beads were obtained from Bangs Laboratories, Inc. (Fisher, IN, USA). EVs derived from A549 non-small cell lung cancer (NSCLC) were purchased from ATCC (Manassas, VA, USA). Dulbecco’s Modified Eagle Medium (DMEM), Antibiotic-Antimycotic (Anti-Anti), and Exosome-depleted Fetal Bovine Serum were purchased from Thermo Fisher Scientific (Waltham, MA, USA). Fetal Bovine Serum (regular) was purchased from Hyclone Laboratories, Inc. (Logan, UT, USA). MagCapture Exosome Isolation Kit PS was purchased from FUJIFILM Wako Pure Chemical Corp. (Richmond, VA, USA). *N,N*′-Bis [4-(4,5-dihydro-1*H*-imidazol-2-yl)phenyl]-3,3′-p-phenylene-bis-acrylamide dihydrochloride (GW4869) was obtained from Cayman CHEMICAL (Ann Arbor, MI, USA). Cell lines, including Huh-7 hepatoblastoma cells, non-small cell lung cancer cells (A549), and breast cancer cells (MDA-MB-231), were purchased from ATCC (Manassas, VA, USA). Materials to build the size-based exosome isolation platform (ExoTIC [25]) were obtained from McMaster-Carr (Los Angeles, CA, USA), Cytiva (Marlborough, MA, USA), and Sterlitech (Kent, WA, USA). Culture media for HUVEC cells were obtained from PromoCell GmbH (Heidelberg, Germany).

### 2.2. Preparation of Nanovesicles

The detailed procedures for synthesizing 100 nm liposomes with different lipid membrane compositions, preparation of EVs from mouse hepatocytes with embedded green fluorescent protein, EVs from human hepatocellular carcinoma (HuH-7), and EVs from HUVEC and MDA-MB-231 cell lines are presented in the Appendix A under Methods.

### 2.3. Device Assembly and Electrical Impedance Measurements

The device consists of two modules: a micropipette-based dielectrophoretic device for entrapment of the vesicles and a digital impedance analyzer setup for in situ impedance measurements of the trapped vesicles (Appendix A). The fabrication procedure of the micropipette-based dielectrophoretic device has been previously reported by our group [23,24,26]. After assembling the device, 10 µL PBS solution and 10 µL PBS solution containing the vesicles at concentration of 6.55 × 10^6^ particles/µL were injected into chambers to immerse the base side and tip side of the pipette respectively. A set of platinum electrodes 0.51 mm in diameter was placed into the chambers for the application of 10 V/cm DC across the pipette for 5 min (Appendix A). The entrapment of the particles was simultaneously observed and recorded using an inverted microscope (TE2000-S, Nikon Instruments, Melville, NY, USA) and a high-resolution camera (Andor NeoZyla 5.5, Oxford Instruments, Abingdon, UK) at a capture rate of 100 frames/second.

A digital impedance analyzer (HF2LI, Zurich Instruments, Zürich, Switzerland) was connected to the second set of platinum electrodes with 130 µm diameter. The electrodes were precisely placed across the trapped particles 20 µm apart via a multi-micromanipulator system (MPC-200, Sutter Instrument Company, Novato, CA, USA). The impedance of the trapped particles was measured as an AC field with a peak amplitude of 100 mV swept from 0.5 MHz to 50 MHz. Frequency-based logarithmic sweep mode was used to record the amplitude and phase of the impedance signal to generate the impedance spectrum (Appendix A). In order to obtain magnitude opacity values at frequencies of interest, we generated a polynomial curve fit of the measured impedance spectrum using the MATLAB function Polyfit. The magnitude opacity values were extracted based on the fitted polynomial function.

For statistical analysis, impedance measurements of each sample were repeated at least 15 times unless otherwise noted and the results were presented as average and standard deviation. Two-sample *t*-tests were performed to compare the two population means, where a *p*-value < 0.05 (**) was considered statistically significant [27].

## 3. Results and Discussion

Studies have shown that the impedance of cells under AC field exhibits variation as a function of frequency. Generally, at a low range of frequency (~kHz), cells are insulating and resisting the current flowing into their interior, and thus the impedance is dominated by the cell’s volume. As the frequency increases (>1 MHz), the cell’s membrane exhibits a capacitive response due to the polarization of the interface between their membrane and the surrounding medium; hence, the impedance is influenced by the membrane capacitance. At higher frequencies (>10 MHz), the electric field (E-field) can penetrate through the cell membrane and polarize the cytoplasm, and thus the impedance reflects the cytosolic conductance of the cell [12]. However, other studies have also shown different frequency responses for phospholipid vesicles with smaller diameters, which reflects their size and surface charge as well as the dielectric properties of their membranes and cytosol [12,28]. Here, we investigated the impedance of a cluster of EVs harvested from different parental cells or cells cultured in different culture conditions at a wide range of frequencies (0.5 MHz to 50 MHz) to detect EVs based on their unique dielectric properties.

### 3.1. Magnitude Opacity

The impedance signal was reported to be influenced by the concentration of entrapped particles [29]. This effect, to some extent, could be compensated by presenting the impedance signal as magnitude opacity, represented in Equation (1). Magnitude opacity O(f) is defined as a ratio of the impedance at all frequencies Z(f) to the impedance $Z\left(f_{reference}\right)$ measured at a size-dependent reference frequency (0.5 MHz) [30]. This concept has been widely applied in cell cytometry to normalize the impedance signal with respect to a cell’s size and its relative position to the electrodes [31,32]. Thus, the opacity, O(f), a volume-independent parameter, would mostly reflect the impedance response of the EVs in terms of their dielectric properties.
(1)$$O\left(f\right)=\frac{Z\left(f\right)}{Z\left(f_{reference}\right)}$$

To verify that the magnitude opacity provides information about the dielectric properties of vesicles, cluster of liposomes at two different concentrations were analyzed. We have previously showed that our device is capable of trapping more vesicles in a form of clusters as the duration of applied E-field increased [24]. Thus, 100 nm liposomes were trapped by applying a 10 V/cm E-field for 2- and 5-min intervals. Microscopic images (Figure 2a) showed that a higher concentration of liposomes was collected after applying the voltage for 5 min (Figure 2(aii)) compared to a 2 min entrapment interval (Figure 2(ai)). The number of trapped liposomes were quantified as 2.2 × 10^6^ for 2 min and 5.4 × 10^6^ for 5 min entrapment after releasing them into 10 µL fresh PBS buffer, followed by nanoparticle tracking analysis (NTA) [23]. The impedances of two clusters were normalized to obtain the magnitude opacity and the results were compared with the impedance of the system without liposomes (before entrapment) (Figure 2b and Appendix A). The results indicated no statistically significant difference (*p* > 0.05) between the two concentrations of entrapped liposomes, while they are significantly different from the impedance of the system without any liposomes. This experiment was repeated with COOH-PS beads and other particles of similar size distributions to validate the magnitude opacity analysis (data not shown). The overlapped magnitude opacity of liposomes at two intervals suggested that the opacity concept can be utilized to mainly analyze the dielectric properties of the vesicles despite their cluster size.

### 3.2. Detection of Nanoparticles with Known Dielectric Properties

To verify the concept of impedance spectroscopy for nano-size particles, liposomes with known dielectric properties were synthesized and measured and a mathematical model was constructed based on an equivalent circuit to support the empirical results (Appendix A). Two sets of 100 nm liposomes were synthesized with different membrane compositions as molar ratios of L-α-phosphatidylcholine (PC) and cholesterol (CH) were changed from 10-to-1 and 1-to-10 ratios, CH:PC_(1:10)_ and CH:PC_(10:1)_, shown in Figure 3a. The capacitance and resistance of the PC lipid bilayer were reported in the literature as 0.38 $\mu F/{\mathrm{cm}}^{2}$, 1.44 $\times \;10^{4}\;\Omega \cdot {\mathrm{cm}}^{2}$, and for the CH bilayer as 0.61 $\mu F/{\mathrm{cm}}^{2}$, 2.12 $\times \;10^{6}\;\Omega \cdot {\mathrm{cm}}^{2}$ by electrochemical impedance spectroscopy [33]. Similar sized COOH-PS beads were selected as reference particles on the basis of their relatively explicit dielectric properties.

The customized mathematical model was built for the cluster of liposomes and COOH-PS beads suspended in PBS buffer, which is described in detail in the Appendix A. In brief, the impedance of the particles (Z_mix_) was estimated by firstly extracting the particles’ permittivity and conductivity based on the capacitance and resistance of the membrane and inner medium to obtain the complex permittivity ${\tilde{\varepsilon }}_{mix}$, followed by estimating Z_mix_ using Maxwell’s mixture equation [12]. The estimated Z_mix_ was implemented into the equivalent circuit to calculate the impedance of the system. The mathematical estimation of particles’ impedance at a wide frequency range was presented as magnitude opacity spectrum. It is important to note that a quantitative comparison of magnitude opacity values between the empirical results and the values obtained from the mathematical model is not exact since the mathematical model has been simplified and the impedance could potentially be influenced by other factors, such as non-ideal characteristics of the measurement electronics [31,34]. Hence, we mainly focus on the comparison between the relative differences in particles’ impedance obtained from empirical and mathematical results, rather than their exact values.

Figure 3b represents the magnitude opacity obtained by the mathematical model of liposomes with different compositions and COOH-PS beads. Due to enriched content of highly resistive cholesterol in liposomes with CH:PC_(10:1)_, higher opacity was obtained when compared to liposomes with CH:PC_(1:10)_ composition. COOH-PS beads have lower magnitude opacity than liposomes, as previously reported by our group, owing to their negatively charged carboxylic acid functional groups [29]. Figure 3c shows the empirical comparison of magnitude opacity for the same particles. A clear difference was also observed empirically for liposomes with different compositions, and the CH:PC_(10:1)_ liposome showed higher magnitude opacity when compared to the CH:PC_(1:10)_ liposomes, which was in agreement with the theoretical model (Figure 3b). These comparisons illustrate that the EIS has the sensitivity to distinguish between a cluster of nanovesicles based on the difference in their membrane compositions, which can be translated to their membrane capacitance and resistance under a wide range of frequencies (10–50 MHz).

### 3.3. Detection of EVs with Different Membrane Compositions

To further test the sensitivity of the system in terms of membrane compositions, the EIS was utilized to measure the impedance of EVs that differ solely in their membrane compositions. To design the experiments, EVs derived from primary hepatocytes were engineered to have green fluorescent protein (GFP+) embedded in their membrane and were compared to the EVs harvested from wildtype hepatocytes lacking the GFP protein (GFP−) [35]. We postulate that the localization of GFP in the membranes of EVs would lead to the alteration of their dielectric properties, which would be detected by EIS. The magnitude opacity spectra of two EVs are shown in Figure 4a. Results showed detectable opacity when GFP− and GFP+ EVs at frequencies higher than 10 MHz (10–50 MHz) were compared, as illustrated in Figure 4b and Appendix A. Although the differences between the opacities of EVs are relatively small here, we have observed consistent results with significant differences (*p* < 0.05) when various batches of EVs were measured (15 trials) at different time points. We believe the sensitivity of the system can be further improved by reducing the dimensions of sensing electrodes at fixed positions in an integrated device in future studies. The lower magnitude opacity of GFP+ EVs compared to the wild type (GFP−) could most likely be due to an increase in the membrane conductivity as a result of the incorporated charged green fluorescent proteins. Based on the constructed equivalent circuit model (Appendix A), the addition of GFP+ in an EV membrane should reduce the resistance of the membrane, resulting in a lower magnitude opacity when compared to the wild type. In addition, the relative opacity of EVs with different membrane compositions is in agreement with the relative opacity spectrum obtained from liposomes with different lipid membrane contents, as described above, suggesting that the system could detect nanovesicles with different membrane compositions at an intermediate to high frequency range.

### 3.4. Detection of EVs Secreted from Cells Treated under Different Culture Conditions

EVs with diverse membrane and cytosolic compositions were selected by harvesting them from human hepatocellular carcinoma cell lines under different culture conditions (Figure 5a). Palmitate acid (PA) a pro-inflammatory fatty acid that can stimulate hepatocytes to generate pro-inflammatory EVs [36] was added to the culture media. PA was also reported to cause variations in EVs’ lipidomic and miRNA expression profiles [37]. Sphingomyelin phosphodiesterase 3 (SMPD 3) specific inhibitor (GW4869) was reported as a neutral inhibitor of sphingomyelinase to attenuate the inflammatory effect in cells [38]. Cells were cultured under the mixture of PA and GW4869 and the harvested EVs were compared to EVs extracted from cells treated with PA. EVs collected from cells under no stimulus were selected as a control. The inflammatory response of EVs collected from these three conditions was examined by culturing EVs with mouse bone marrow-derived macrophages (BMDM) which were then analyzed for the cytokines interleukin 6 (IL-6) and tumor necrosis factor alpha (TNF-α) mRNA expression levels via quantitative polymerase chain reaction (q-PCR) (Figure 5b). Results showed significantly elevated expression levels of TNF-α and IL-6 mRNAs, reflecting the inflammatory responses of EVs derived from the PA-treated culture condition. GW4869 inhibited the inflammatory effect caused by PA, and thus EVs harvested from cells treated with the mixture of PA and GW4869 resulted in a reduction of mRNA expression levels of TNF-α and IL-6.

Given the potential variations in the biochemical properties of EVs harvested from cells under the pro-inflammatory stimulus, EV dielectric properties were studied by EIS (Figure 5c,d and Appendix A). Figure 5c illustrates the magnitude opacity spectrum of EVs at a wide range of frequencies (1–50 MHz). EVs harvested from cells treated with PA showed higher magnitude opacity from 1–15 MHz when compared to the control EVs harvested from cells treated with no PA or GW4869. This could potentially be due to the increase of ceramide lipids in the EVs’ membrane composition under PA-tread condition [39]. Since the capacitance of the ceramide lipid bilayer is lower than the phosphatidylcholine bilayer [40], EVs containing a higher concentration of ceramide lipid will have a lower membrane capacitance, resulting in higher magnitude opacity when compared to the control EVs (red line). Additionally, the opacities of both EVs (PA) and EVs (PA + GW) showed sudden decreases at 15 MHz when compared to the control EVs which showed a more linear drop in opacity; this observation could potentially be the signature of EV membrane lipids and hence the membrane capacitance in the case of EVs (PA) and EVs (PA + GW). However, as frequency increased to 15 MHz, the opacity of EVs treated with PA + GW increased initially and showed similar characteristics to the control EVs at intervals of 15–50 MHz, while PA-treated EVs’ opacities continued to drop linearly. This interesting observation could potentially illustrate the inhibitory effect of GW4860 on PA-treated cells and consequently on their secreted EVs, as the opacity spectrum of control EVs and EVs (PA + GW) showed similar patterns. In addition, from ~35 MHz to 50 MHz, the opacity of PA-treated EVs dropped at a faster rate when compared to the control EVs and EVs (PA + GW) with more plateaued opacities. This fast drop rate in opacity at the higher frequency range for PA-treated EVs could be associated with the changes in their cytosolic contents, hence the overexpression of RNA content in their lumen [41], which could lead to the reduction of cytosolic resistance and magnitude opacity when compared to the control EVs and EVs (PA + GW) [37].

### 3.5. Differentiating EVs from Lipoproteins

We further utilized the EIS to discriminate EVs from lipoproteins which share similar properties in their biophysics but have different biochemical properties. Lipoproteins have single-layer phospholipids embedded with apolipoproteins and are in charge of the transportation of water-insoluble hydrophobic lipid molecules into extracellular fluids [42]. Although both lipoproteins and EVs have embedded proteins in their membrane structure, studies have shown that they have diverse lipid and membrane protein compositions [43], which could potentially lead to variations of their dielectric properties. In addition, lipoproteins encapsulate hydrophobic lipid molecules, including triglycerides (TGs) and cholesterol in their lumen, while EVs have high concentrations of charged proteins. The impedance of EVs derived from A549 NSCLC cells and very low-density (VLD) lipoproteins from human plasma was measured under a wide frequency spectrum (Figure 6a). The difference between their opacity became significant at frequencies above 10 MHz (10–50 MHz), which is illustrated in Figure 6b and Appendix A. The opacity of EVs was lower than lipoproteins in all frequencies, which could be attributed to the higher concentration of overall charged molecules, including proteins and nucleic acids embedded in their membrane and lumen. At the interval between 1–30 MHz, the opacity of both vesicles dropped linearly, with an increased drop rate from 30–40 MHz. However, in the 40–50 MHz interval, the opacity increased at 45 MHz, followed by a decrease at 50 MHz. The fluctuation patterns in opacity at a higher range of frequency could potentially be correlated with the penetration of the electric field to the vesicles’ lumens and their corresponding cytosolic conductance. However, to precisely correlate the membrane and cytosolic composition of nanovesicles with their frequency-dependent impedance, precise molecular analyses of vesicles, such as proteomic, lipidomic, and genomic, would need to be performed. This will be the subject of our future studies.

### 3.6. Detection of EVs Derived from Different Cellular Origins

Detection of the dielectric properties of EVs harvested from different cellular origins is of particular interest since the secreted EVs could provide essential biochemical information, including nucleic acid and protein contents that were inherited from the parental cells [44]. Here, we utilized EVs harvested from two common cell lines, umbilical vein endothelial cells (HUVECs) and epithelial human breast cancer cells (MDA-MB-231), to investigate their differences by EIS. EVs secreted from MDA-MB-231 cells are widely studied for the enriched oncogenes in the lumen which lead to oncogenic transformation [45]. HUVEC cell lines were commonly used to study the role of angiogenic EVs secreted from MDA-MB-231 cell lines in tumor growth and metastasis [46]. When the impedance of EVs derived from these two cell lines were compared, significantly higher magnitude opacities were observed for MDA-MB-231-derived EVs at frequencies of 10 MHz to 20 MHz (Figure 7a and Appendix A). However, as the frequency increased above 20 MHz, the difference between their opacities became insignificant (at 30 MHz and 40 MHz), and as the frequency reached 50 MHz, the opacity of EVs derived from HUVECs exceeded the EVs harvested from MDA-MB-231 cells. Given the previous observations, we postulated that the shift in magnitude opacity at frequencies above 30 MHz could potentially be caused by a dominant effect of cytosolic conductance in EVs which as a result overturned the difference in opacity caused by their membrane capacitance. Although these initial observations provide an insight with regard to cytosolic and membrane effects on EVs’ dielectric properties at different ranges of the frequency spectrum, more comprehensive and precise studies of EVs’ molecular profiles need to be conducted to correlate the exact role of membrane and cytosol with their frequency-dependent dielectric properties.

### 3.7. Detection of EVs with Different Size Distributions

Besides the effect of parental cells on EVs biochemical and biophysical characteristics, the heterogeneity of EVs in their size also adds to the complexity of their characterization [47]. It has been reported that EVs have different biochemical properties, including protein, lipid, and nucleic acid contents, at different size ranges [48,49]. For instance, Zhang et al. showed that EVs derived from an MDA-MB-231 cell line at different size ranges have different biochemical and biophysical properties, including zeta potential, stiffness, lipid composition, and proteomic and nucleic acid payload [48]. Thus, we measured the impedance of EVs derived from an MDA-MB-231 cell line at different size distributions to investigate the correlation between EVs’ size and their dielectric properties. EVs derived from MDA-MB-231 cells were isolated utilizing a size-based sorting platform, ExoTIC, developed by our group [25]. Figure 7(bi) shows the magnitude opacity spectrum of EVs at different size ranges, and the results illustrated significant differences between each group of EVs (Figure 7(bii,biii) and Appendix A). An interesting pattern in this set of data has been observed in which the magnitude opacity increased as the EVs’ size distribution increased, for instance, a lower magnitude opacity obtained for EVs with a 30–50 nm size range when compared to EVs with a 50–80 nm range, and so on. This could potentially be explained as a result of the relatively higher ratio of charged molecules, including nucleic acids and proteins, to the ratio of inert lipid bilayer membrane in EVs of smaller size when compared to EVs of a larger size. Although this preliminary data provides important information with regards to the correlation of EVs’ size distribution and their dielectric properties, it is not feasible to report the exact causes of their impedance differences given the fact that each subpopulation is different in more than one biophysical and/or biochemical parameter. Thus, the EIS can be utilized as a tool to provide a rapid detection of EVs of different sizes, and a comprehensive downstream analysis of EVs’ molecular profiles will be required to further study the effects of their membrane or cytosolic cargos.

## 4. Conclusions

In summary, this study has reported a label-free biosensor for detection of EVs based on their unique dielectric properties. The system consisted of a micropipette-based dielectrophoretic device integrated with an EIS to measure the impedance of immobilized vesicles at a wide range of frequencies. The detection principle was mathematically modeled based on an equivalent circuit and was in agreement with empirical results when nanovesicles with known dielectric properties were tested. In addition, the system showed that EVs could be discriminated from lipoproteins which shared similar biophysical properties but differed in their biochemical compositions. Moreover, the system showed sensitivity for detecting EVs with different membrane compositions but the same cytosolic contents at a wide frequency spectrum (10–50 MHz). In addition, the impedance of EVs harvested from cells in different culture conditions and thus different functionality in terms of pro-inflammatory effect were detected at intermediate and high frequency ranges (10 MHz to 50 MHz).

Furthermore, the sensor could detect EVs derived from different cellular origins, which could be further utilized to rapidly characterize EVs in diagnostic and therapeutic applications. We also illustrated the capability of the EIS to differentiate EVs at different size distributions, which presented the heterogeneity of their dielectric properties associated with their biochemical properties. Overall, this novel biosensor opens up a new way for rapid, label-free, and non-invasive characterization of a cluster of EVs (~1 million nanovesicles) based on their unique dielectric properties which can be associated with their charge-dependent membrane and cytosolic molecular contents. This technique also has great potential to be further evolved as a diagnostic tool for the detection of pathogenic EVs and can be applied for monitoring EV cargos in personalized therapeutics.

## Figures and Tables

**Figure 1 biosensors-12-00104-f001:**
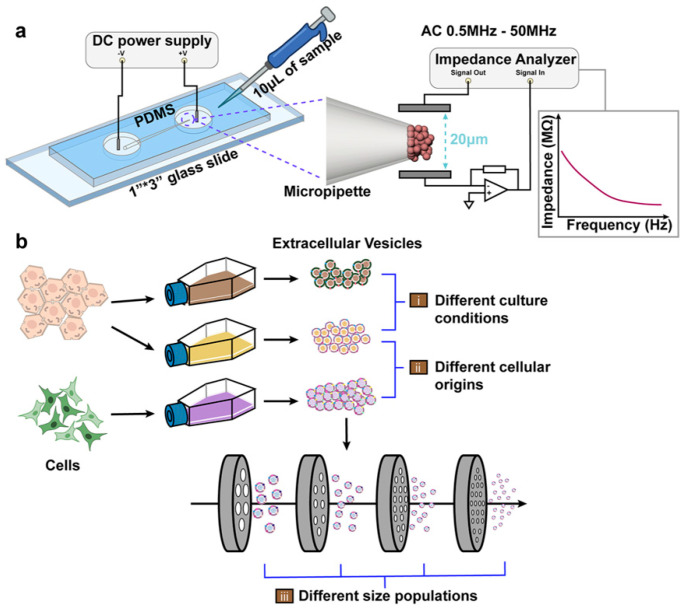
(**a**) Schematic of an integrated iDEP and EIS system. The setup was composed of a borosilicate micropipette placed in between two PDMS chambers. DC bias was applied to trap vesicles at the pipette’s tip by electrokinetic forces, followed by measuring of the impedance of the collected vesicles, utilizing the sensing electrodes, at a wide frequency spectrum (0.5 MHz to 50 MHz). (**b**) Detection of EVs harvested from: (**i**) cells under different culture conditions, (**ii**) different cellular origins, and (**iii**) populations of different sizes.

**Figure 2 biosensors-12-00104-f002:**
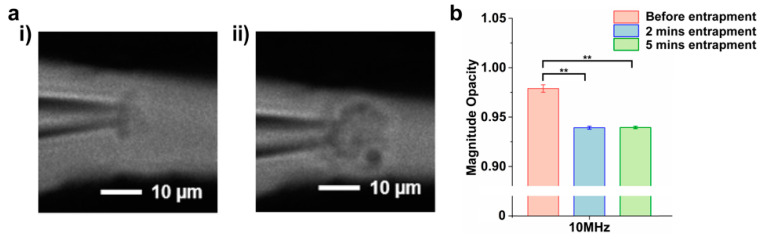
(**a**) Microscopic images of liposome entrapment for (**i**) 2 min, with the average number of trapped vesicles as 2.2 × 10^6^, and (**ii**) 5 min, with the average number of trapped vesicles as 5.4 × 10^6^. (**b**) The magnitude opacity comparison among empty pipette (before entrapment) and liposome clusters extracted at two different time intervals at 10 MHz. (** *p* < 0.05, n = 3).

**Figure 3 biosensors-12-00104-f003:**
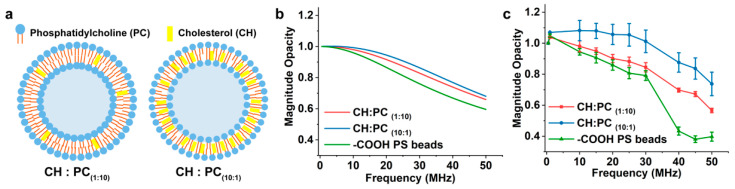
(**a**) Schematic representations of two types of liposomes with different membrane compositions (CH:PC_(1:10)_ and CH:PC_(10:1)_). The relative magnitude opacity of liposomes and COOH-PS beads (the average size of all particles is 100 nm) utilizing (**b**) the mathematical model and (**c**) the experimental measurement. (n = 15).

**Figure 4 biosensors-12-00104-f004:**
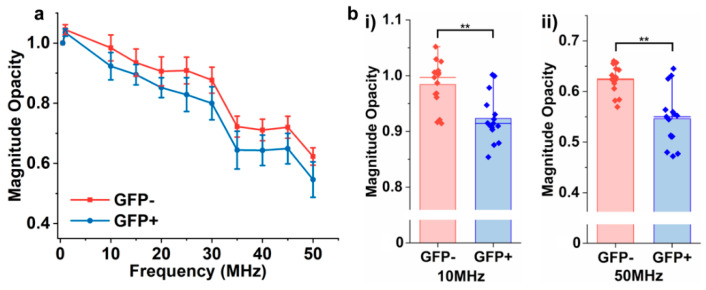
(**a**) The magnitude opacity spectrum of EVs derived from wildtype primary hepatocytes (GFP−) and GPF+ hepatocytes. (**b**) Significant differences in magnitude opacity were observed at 10 MHz and higher frequencies up to 50 MHz. (** *p* < 0.05, n = 15).

**Figure 5 biosensors-12-00104-f005:**
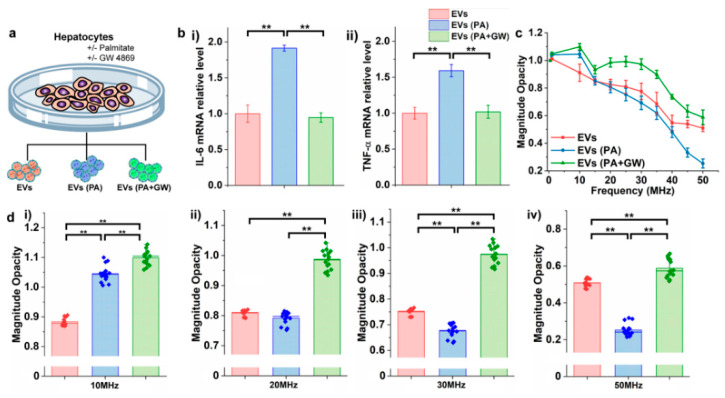
(**a**) Isolation of EVs from human hepatocellular carcinoma cell lines under normal culture medium, the PA-treated condition, and the mixture of PA- and GW4869-treated conditions. (**b**) Mouse bone marrow-derived macrophages (BMDMs) were cultured with EVs for mRNA expression analysis by the quantitative polymerase chain reaction of (**i**) IL-6 mRNA and (**ii**) TNF-α mRNA. (**c**) Magnitude opacity spectrum of EVs derived from human hepatocellular carcinoma cell lines under three culture conditions. (**d**) Box plots of magnitude opacity comparison of EVs at three culture conditions at 10–50 MHz. (** *p* < 0.05, n = 15).

**Figure 6 biosensors-12-00104-f006:**
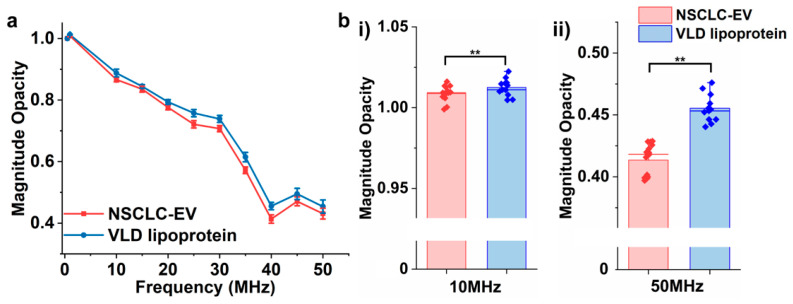
(**a**) Magnitude opacity spectrum of EVs derived from A549 non-small cell lung cancer (NSCLC) cell line and very low-density (VLD) lipoprotein. (**b**) Bar plots of magnitude opacity comparison of NSCLC and VLD lipoproteins at 10 MHz and 50 MHz. (** *p* < 0.05, n = 12).

**Figure 7 biosensors-12-00104-f007:**
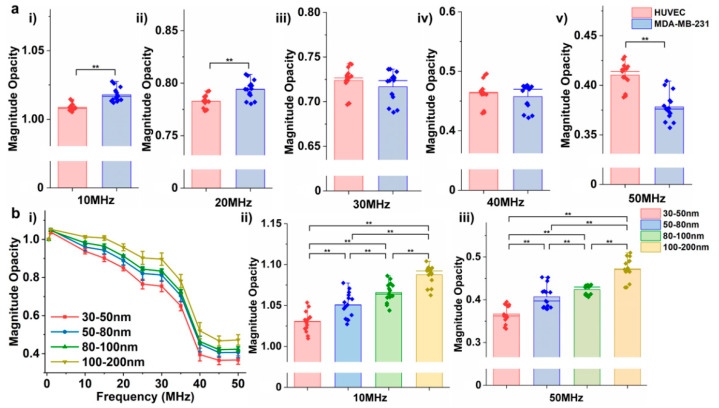
The magnitude opacity comparison of EVs extracted from (**a**) human umbilical vein endothelial cells (HUVECs) and epithelial human breast cancer (MDA-MB-231) cells. (** *p* < 0.05, n = 15.) (**b**) (**i**) Magnitude opacity spectrum of EVs at different size ranges isolated from an MDA-MB-231 cell line by ExoTIC. (**ii**,**iii**) Bar plots of magnitude opacity comparison of four EVs subsets at 10 MHz and 50 MHz. (** *p* < 0.05, n = 15).

## Data Availability

Not applicable.

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
