# Peer review of "A Label-Free Electrical Impedance Spectroscopy for Detection of Clusters of Extracellular Vesicles Based on Their Unique Dielectric Properties"

_biosensors, 2022, doi:10.3390/bios12020104_

Round 1

Reviewer 1 Report

Esfandiari and coauthors reported a frequency-dependent impedance measurement system to characterize EVs based on their unique dielectric properties. Generally, this is a well organized paper. However, the current version still owns some flaws that needs to be addressed before publication.

1, for the title, the authors mentioned the detection of EV, however, the samples are actually a cluster of EVs, which could mislead the readers;

2, actually, the introduction is quite nicely written, giving  quite a clear background for this research and the innovation here. However, like Q1, the discussion are focused on single EV detection, not a cluster, so, could the authors revise it accordingly? To make the authors understand why the detection of clusters are important. And also, if yes, some more relevant papers on the detection of cell clusters or EV clusters should be well cited and discussed.

3, for the setup of the system, the reviewer can not understand where the electrodes are, and how to measure the impedance, could the authors provide a photo of the system?

4, for figure 5d, when the frequency changed, the detection signals also changed, how to quantitatively provide conclusion? Actually, it is hard for the reviewer to understand how this system could work quantitatively, with such a poor selectivity.

5, although the authors mentioned that the impedance could be transferred into magnitude opacity,  however, some initial impedance data should be provided for reviewers to understand, such as shown in SI.

6, most of the references are quite old, which should be updated with papers published in the past 3 years.

7, this system could only used for detection of known targets, which would limits its potential applicaiton. 

8, for a biosensor, the sensitivity, selectivity should be well discussed in the paper.

Author Response

Thank you for valuable suggestions. We have revised the manuscript and supporting information accordingly. 

Reviewer 2 Report

The article “A label-free electrical impedance spectroscopy for detection of small 

extracellular vesicles based on their unique dielectric properties” presents the development of a frequency-dependant impedance measurement system to characterize extracellular vesicles. The device is composed of an iDEP device for trapping and immobilizing and an impedance sensor for characterization. Different extracellular vesicles are compared with different membrane compositions, secreted from cells with different conditions, from different origins and size distributions. The article is well written and comprehensive. I have a few minor remarks that should be addressed prior to publications.

Detailed comments:

  • Figure 1: reindicate the dimensions involved in the device (distance between the electrodes, between the chambers, diameter of the chambers etc)
  • L147: PBS, which is very conductive, is used for the experiment together with a relatively high voltage in DC. Please comment on the influence of such voltage in the experiment (Joule heating, effect on cells).
  • L161: the technique used with the generation of a polynomial curve fit is a bit unclear. Please explain why a fit is used and not directly the measured values.
  • L200: How many repeats were done for this experiment? It seems like n=1 but in Figure 2 the “average number” is mentioned which makes this very confusing.
  • L207: the experiment with COOH-PS in mentioned but without neither the results nor the conclusion.
  • The reference frequency used for the opacity should be indicated in the legend of each figure.
  • Figure 4: why is the magnitude opacity at 0.5 MHz not always 1 as it should be by definition (This comment is valid for figure 3c, 5c, 6a, 7b)
  • Apart from the error bars of Figure 5b, all the other ones in the article seem to be missing the lower part. It should be corrected!

Minor remarks:

  • Be consistent with the unit format and keep using a space between a number and its unit.
  • L265: typo: should be alteration instead of alternation
  • L338: VLD is defined but not VLDL

Author Response

Thank you for your valuable suggestions. We revised the manuscript and supporting information accordingly. 

Round 2

Reviewer 1 Report

The reviewer would appreciate the revisions provided by the authors. However, by carefully checking this paper, and especially comparing with refs. 21-22, the reviewer feels hard to provide a more positive recommendation.

1, for my previous questions, regarding to sensitivity and selectivity, the current work did not provide any answers, but leaving for future works;

2, for the detection of different cells, there is no analysis why these cells could be distinguished;

3, it makes no sense of detecting a cluster of cells, by such a complicated  but inaccurate system (discussed in previous questions already, such as the increase of MO with frequency).

Based on these major problems, I do not suggest the publication of this current work, unless the sensitivity and selectivity are realized.

Author Response

Thank you for pressing on these important points. We have addressed the comments throughout the manuscript. please see attached file for point by point responses. 
